# An explanation for origin unwinding in eukaryotes

**Lance D Langston[1,2], Michael E O'Donnell[1,2]***

[1]The Rockefeller University, New York, United States; [2]Howard Hughes Medical Institute, New York, United States

**Abstract** Twin CMG complexes are assembled head-to-head around duplex DNA at eukaryotic origins of replication. Mcm10 activates CMGs to form helicases that encircle single-strand (ss) DNA and initiate bidirectional forks. How the CMGs melt duplex DNA while encircling it is unknown. Here we show that *S. cerevisiae* CMG tracks with force while encircling double-stranded (ds) DNA and that in the presence of Mcm10 the CMG melts long blocks of dsDNA while it encircles dsDNA. We demonstrate that CMG tracks mainly on the 3'−5' strand during duplex translocation, predicting that head-to-head CMGs at an origin exert force on opposite strands. Accordingly, we show that CMGs that encircle double strand DNA in a head-to-head orientation melt the duplex in an Mcm10-dependent reaction.
DOI: https://doi.org/10.7554/eLife.46515.001

## Introduction

Cellular replicative DNA helicases are hexameric ring-shaped motors that track on single-strand (ss) DNA, acting as a wedge to unwind the duplex (*Lyubimov et al., 2011*). In bacteria, the circular helicase is a homohexamer that directly assembles onto an AT-rich ssDNA region melted by initiator proteins, but in eukaryotes the origin recognition complex (ORC) does not unwind DNA (*Bleichert et al., 2017*). Instead, in G1-phase, two Mcm2-7 hexamer rings are assembled head-to-head (N-to-N) around double-strand (ds) DNA by ORC, Cdt1 and Cdc6 (*Bell and Labib, 2016*; *Evrin et al., 2009*; *O'Donnell et al., 2013*; *Remus et al., 2009*). In S-phase, additional factors assemble Cdc45 and GINS onto each Mcm2-7 to form two replicative helicases on dsDNA oriented head-to-head (*Ilves et al., 2010*; *Moyer et al., 2006*; *Parker et al., 2017*). Each helicase contains 11 different subunits, and the term CMG (Cdc45, Mcm2-7, GINS) was coined because it contains Cdc45, Mcm2-7, and the four subunit GINS complex (*Ilves et al., 2010*; *Moyer et al., 2006*; *Parker et al., 2017*). Mcm10 is required to activate each CMG to encircle ssDNA for bidirectional replication (*Douglas et al., 2018*; *Heller et al., 2011*; *Kanke et al., 2012*; *van Deursen et al., 2012*; *Watase et al., 2012*; *Yeeles et al., 2015*). However, the detailed processes of origin melting and the transition of CMG from dsDNA to ssDNA translocation are unknown. In particular, it is not clear whether CMG and Mcm10 alone are sufficient to mediate this transition or whether other factors required for CMG assembly are also involved.

Recent electron microscopy (EM) studies of *Saccharomyces cerevisiae* (S.c.) CMG at a replication fork show that CMG tracks on ssDNA with its N-tier facing the fork (*Douglas et al., 2018*; *Georgescu et al., 2017*). N-first movement of CMG implies that head-to-head CMGs at the origin are directed toward one another when assembled around duplex DNA. Thus, each CMG blocks the forward movement of the other, and the duplex must be unwound a sufficient distance for both CMGs to expel one strand for their transition from dsDNA to ssDNA, which they must do to pass one another and form bidirectional forks (*Figure 1*). These last steps of origin initiation require the Mcm10 protein (*Douglas et al., 2018*; *Kanke et al., 2012*; *van Deursen et al., 2012*; *Watase et al., 2012*; *Yeeles et al., 2015*). Mcm10 contains a DNA binding region (*Warren et al., 2008*) and

***For correspondence:**
odonnel@rockefeller.edu

**Competing interests:** The authors declare that no competing interests exist.

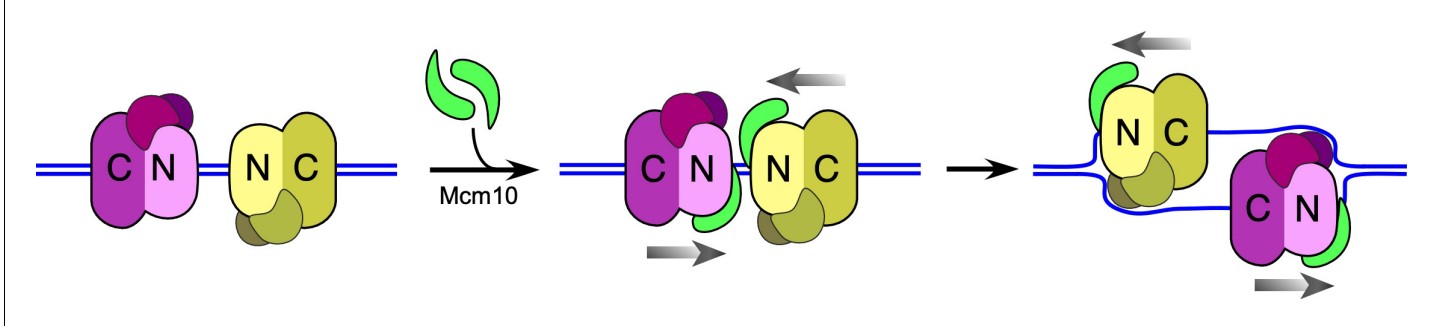

**Figure 1.** CMG action at origins. CMG complexes, which appear as stacked rings with separate N- and C-domains, translocate in the N-first direction. Initially, two N-to-N (head-to-head) CMGs (purple and yellow) are assembled surrounding dsDNA (blue lines) at origins (left), and the tight connection of the inactive Mcm2-7 double hexamer is lost (*Abid Ali and Costa, 2016*; *Costa et al., 2014*), depicted as a separation between the CMGs (left). Upon addition of Mcm10 (green, middle) both CMGs switch to opposite strands of ssDNA and can pass one another to form bidirectional forks (right).
DOI: https://doi.org/10.7554/eLife.46515.002

interaction with the N-terminal regions of Mcm2 and Mcm6 has been demonstrated (*Douglas and Diffley, 2016*; *Lõoke et al., 2017*). Recent cross-linking/mass spectrometry studies reveal a large surface area of CMG-Mcm10 interaction that includes 6 of the 11 CMG subunits, mostly at the N-surface of CMG but also many sites at the edges and C-surface of CMG (*Mayle et al., 2019*).

The mechanism of CMG + Mcm10 (CMGM) function in the final step of origin initiation has been difficult to probe experimentally. Recent studies of CMG formation at origins demonstrate a disruption of tight connections between the Mcm2-7 hexamers upon maturation to CMGs and untwisting of ~0.7 turns of the double helix per CMG (*Douglas et al., 2018*). However, even if the observed untwisting was due to actual DNA melting, it is too little for CMG to transition to ssDNA (*Douglas et al., 2018*). Thus, the mechanism of how CMG opens the duplex a sufficient length to transition to ssDNA and the role of Mcm10 in this process remain open questions. In this report we use recombinant CMG and Mcm10 and find that, together, they produce sufficient force to open several turns of the duplex while encircling dsDNA, a sufficient length that would enable CMGs to transition onto ssDNA. Hence, initial unwinding of DNA by CMG-Mcm10 can occur without the participation of other replication initiation factors.

## Results

To investigate the activity of the CMG motor while encircling dsDNA, we designed three T-shaped DNA substrates that block passive sliding over the two non-homologous arms at the T-junction (*Figure 2A* and *Figure 2—figure supplement 1*). If CMG is not passive, but instead CMG motors on dsDNA with force, it may melt the T-junction arms and produce unwound products, provided the arms are not too long for CMG to melt. Similar substrates have been used to study ring-shaped helicases that track on dsDNA with force including *E. coli* DnaB helicase and the FtsK chromosome segregation motor (*Bigot et al., 2005*; *Crozat and Grainge, 2010*; *Kaplan and O'Donnell, 2004*; *Lyubimov et al., 2011*). The T-substrates contain a 3' ssDNA dT tail that is required for CMG loading (*Figure 2a* and *Figure 2—figure supplement 2*) followed by a flush ss/dsDNA junction that enables CMG to move onto dsDNA without unwinding it (*Kang et al., 2012*; *Langston and O'Donnell, 2017*) after which it encounters the T-junction block of non-homologous arms (*Figure 2a*). The three T-shaped DNAs are identical except for the length of the branched arms (20, 30 or 60 bp). The non-homologous arms are annealed to a [32]P-radiolabeled 'C' oligo. Proceeding through the T-arm blocks requires that CMG translocates with force while surrounding dsDNA to simultaneously melt both arms, so a T-substrate with 30 bp arms requires melting a total of 60 bp to release the [32]P-C oligo.

CMG was pre-incubated with the T-substrates for 10' and then ATP was added to initiate unwinding as illustrated in *Figure 2A*. An excess of unlabeled C oligo was added 1' later as a trap to prevent re-annealing of any unwound [32]P-C oligo. CMG has only weak activity in melting the 20 bp arms and cannot destabilize the 30 bp or 60 bp arms (*Figure 2B,D*). However, upon adding Mcm10, CMG unwound all three T-substrates (*Figure 2C,D*). Mcm10 alone has no activity in these assays

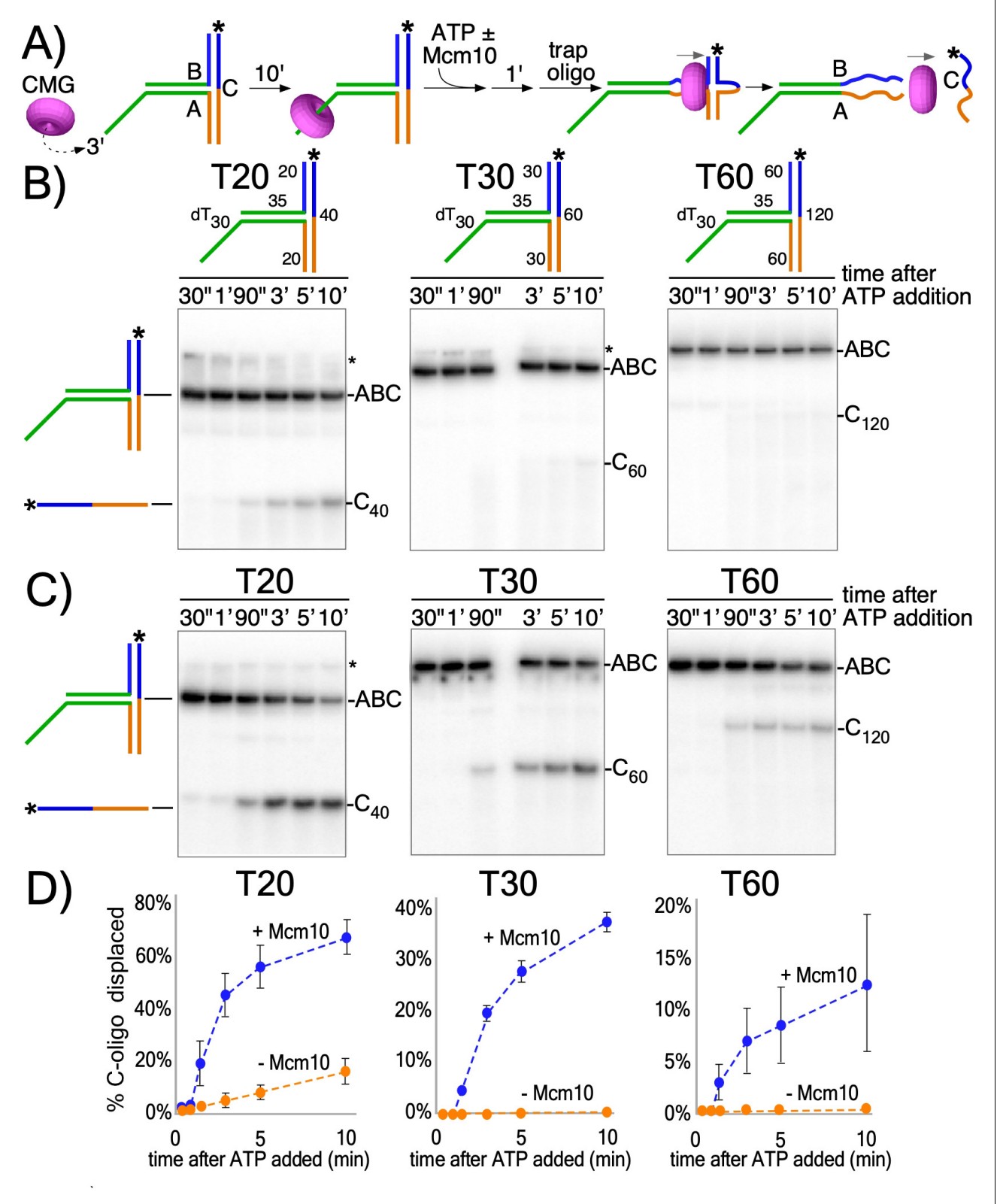

**Figure 2.** CMG translocates with force while encircling duplex DNA. (A) Schematic of reactions using T-substrates with a 3' $dT_{30}$ ssDNA tail for CMG loading, a flush ss/ds junction and 35 bp of dsDNA (green) preceding the non-homologous arm blocks (blue and orange). See *Figure 2—figure supplement 1* for details on the substrates. Unwinding the $^{32}P$-cross-bar oligo (labeled with a *) requires CMG to track with force while encircling dsDNA. (B) Native PAGE gel of time course reactions using CMG (no Mcm10) on T-substrates with different length arms. (C) Repeat of (B) with addition

*Figure 2 continued on next page*

*Figure 2 continued*

of Mcm10. (**D**) Plots of the data from (**B**) and (**C**). Values shown are the average of three independent experiments and the error bars show one standard deviation. The asterisks to the right of the gels indicate gel-shift of the substrate by CMG and Mcm10. See also *Figure 2—figure supplements 2–3*.

DOI: https://doi.org/10.7554/eLife.46515.003

The following figure supplements are available for figure 2:

**Figure supplement 1.** The T-shaped substrates shown in schematic form.

DOI: https://doi.org/10.7554/eLife.46515.004

**Figure supplement 2.** The 3' $dT_{30}$ tail is required for CMG loading onto the T-substrates.

DOI: https://doi.org/10.7554/eLife.46515.005

**Figure supplement 3.** Mcm10 does not unwind the T-substrates without CMG.

DOI: https://doi.org/10.7554/eLife.46515.006

(*Figure 2—figure supplement 3*). The results are consistent with prior studies showing Mcm10 stimulates CMG helicase activity, enhances CMG processivity during unwinding (*Langston et al., 2017*; *Lõoke et al., 2017*), and is required to activate CMG unwinding at origins (*Douglas et al., 2018*; *Kanke et al., 2012*; *van Deursen et al., 2012*; *Watase et al., 2012*; *Yeeles et al., 2015*). Possible roles of Mcm10 in promoting CMG unwinding while encircling dsDNA are presented in the Discussion. The extent of unwinding decreases as the length of the non-homologous arms increases, suggesting either limited processivity of CMG + Mcm10 in directional dsDNA translocation or that CMG expels the non-tracking strand and transitions to ssDNA translocation while unwinding the longer T-substrates; strand expulsion would result in only partial unwinding followed by rapid intramolecular reannealing to reform the starting substrate, a topic we examine later.

Previous studies of multi-subunit, ring-shaped motors that surround and track on dsDNA show that they primarily contact one strand of the duplex. Two of the most well-studied examples are the phi29 DNA packaging motor, which tracks on only one strand while encircling dsDNA (*Aathavan et al., 2009*) and the bacterial clamp loader that also surrounds dsDNA but contacts only one strand (*Simonetta et al., 2009*) (*Figure 3—figure supplement 1*). Similarly, the cryoEM structure of CMG on forked DNA shows that the translocation strand adopts a B-form spiral shape in the motor domains (*Figure 3—figure supplement 1*), implying that the motors need not alter conformation to encircle dsDNA (*Abid Ali et al., 2016*; *Georgescu et al., 2017*). A corresponding spiral of ssDNA in the central channel is a common feature of all replicative helicase structures examined thus far (*Gao et al., 2019*; *O'Donnell and Li, 2018*; *Parker et al., 2017*). These findings suggest CMG may track on one strand while encircling dsDNA. Translocases bind the negative phosphate backbone for mobility (*Kaplan and O'Donnell, 2004*; *Lyubimov et al., 2011*; *Parker et al., 2017*), and thus to test CMG for tracking on one strand of dsDNA we placed ten neutral methylphosphonate linkages on either strand of the duplex preceding the T-substrate arms (*Figure 3A* and *Figure 3—figure supplement 2*), a strategy used to identify the tracking strand for the phi29 motor (*Aathavan et al., 2009*). When the methylphosphonate linkages were on the 3'−5' strand in the direction of unwinding, duplex unwinding was significantly reduced compared to methylphosphonate linkages on the opposite strand (*Figure 3B,C*). This result is consistent with CMG tracking mainly on the 3'−5' strand while surrounding dsDNA, which is also the tracking strand for helicase activity while encircling ssDNA. This is noteworthy given that in the inactive Mcm2-7 double hexamer, each ring makes an equal number of contacts with the 3'−5' and 5'−3' strands (*Abid Ali et al., 2017*; *Noguchi et al., 2017*). Hence, duplex DNA interaction by the MCM and by the CMG is fundamentally different.

The studies of *Figures 2* and *3* show that CMG + Mcm10 tracks on dsDNA with force and interacts with the 3'−5' strand more than the 5'−3' strand. On the basis of these observations, we hypothesized that head-to-head CMGs encircling dsDNA and pushing against one another at origins would pull on opposite strands of the duplex, providing the force necessary to melt the origin. To test this, we designed a linear duplex with CMG loading sites at either end to simulate an origin in which two CMGs are oriented head-to-head on dsDNA (*Figure 4A*). The substrate supports loading of two (at least) oppositely oriented CMG complexes onto duplex DNA that will translocate toward one another and collide in the same way that CMGs do at an origin but without the need for CMG helicase assembly by multiple initiation factors (*Figure 4A*, middle). This allows us to determine if

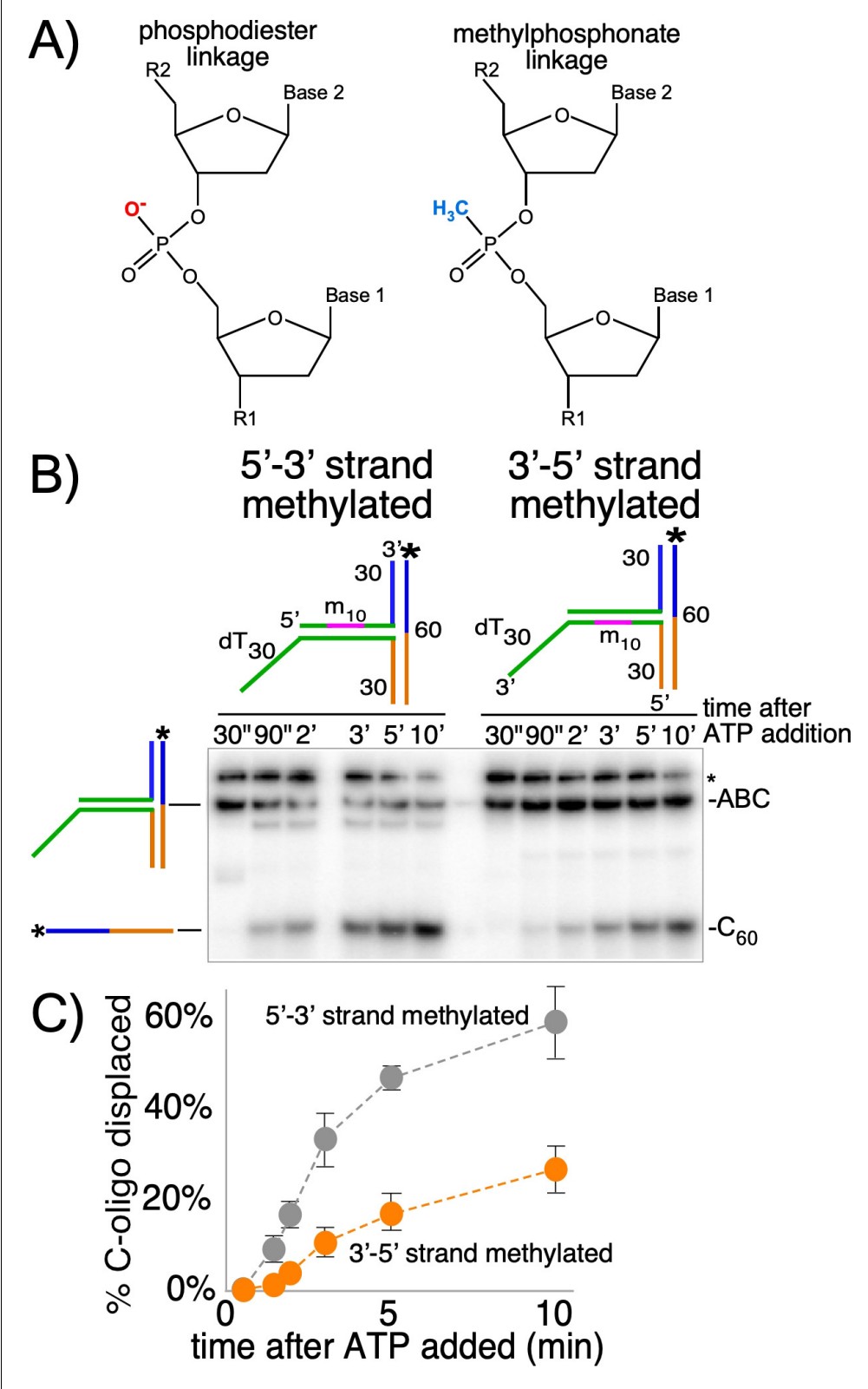

**Figure 3.** CMG+Mcm10 strand interactions during duplex translocation. (A) Diagram of the naturally occurring negatively charged phosphodiester linkage in the phosphate backbone of a DNA chain (left) and the uncharged methylphosphonate linkages used in these experiments (right). (B) The experiment from *Figure 2C* (CMG+Mcm10) was repeated using a T30 substrate with 10 neutral methylphosphonate linkages (shown in pink in the schematics above the gel) in the 5'-3' strand (left) or in the 3'-5' strand (right) of the duplex. See *Figure 3—figure supplement 2* and

*Figure 3 continued on next page*

*Figure 3 continued*

*Supplementary file 1* for further details on the substrate. The * next to the gels indicates gel-shift of the substrate by CMG+Mcm10. (C) A plot of the data from (B) shows the averages of three independent trials using these substrates. The error bars show the standard deviations. Also see *Figure 3—figure supplement 1*.

DOI: https://doi.org/10.7554/eLife.46515.007

The following figure supplements are available for figure 3:

**Figure supplement 1.** Duplex strand ATPases contact only one strand while encircling duplex DNA.

DOI: https://doi.org/10.7554/eLife.46515.008

**Figure supplement 2.** Design of substrates with methylphosphonate linkages in *Figure 3*.

DOI: https://doi.org/10.7554/eLife.46515.009

the duplex melting activity required at origins is inherent in head-to-head CMG + Mcm10 complexes or if additional factors are needed. In this assay, if the substrate is unwound, the trap oligo prevents reannealing by forming a fork with the unwound $^{32}$P-labeled strand and the fork structure migrates more slowly in the gel than the initial substrate (*Figure 4A*, bottom).

Considering that circular helicases like CMG are inefficient in threading onto DNA, we assume that binding of two CMG to a single substrate is probably a rare event. To further reduce the probability of >2 CMG loading, we minimized the opportunity for CMG to bind to the substrate as follows. First, CMG was mixed with the substrate on ice and then incubated for 1 min at 30℃ in the absence of ATP to allow the mixture to reach the reaction temperature. Next, ATP was added to initiate CMG loading and duplex translocation along with Mcm10 to initiate duplex melting (*Douglas et al., 2018*; *Kanke et al., 2012*; *van Deursen et al., 2012*; *Watase et al., 2012*; *Yeeles et al., 2015*). Then 45 s after adding ATP and Mcm10, we added the trap oligo which quenches further CMG loading (*Figure 4—figure supplement 1*). As shown in *Figure 4B* (lanes 1–5), CMG readily unwinds the two-tailed substrate in the presence of Mcm10. Moreover, unwinding continues well beyond the time of addition of the trap, thereby excluding the possibility that additional CMG loading events are required for unwinding (see *Figure 4B and C*; also see *Figure 4—figure supplement 1* for effectiveness of the trap). The reaction is absolutely dependent on addition of Mcm10 (*Figure 4—figure supplement 2*) and also requires ATP and CMG (*Figure 4—figure supplement 3*), showing that the observed unwinding is the product of CMG's ATP-dependent motor activity coupled to Mcm10 function rather than the product of either CMG or Mcm10 acting separately. The reaction also requires loading of head-to-head CMG complexes because duplexes with only a single CMG loading tail at one end or the other were essentially inactive compared to the two-tailed substrate (*Figure 4B*, lanes 6–15 and *Figure 4C*). In overview, these results demonstrate that head-to-head CMG complexes, along with Mcm10, can perform the DNA melting required at origins of replication without the participation of any other factors. Furthermore, the results are consistent with the action of two independent CMGs colliding with one another on dsDNA, in contrast to the highly interconnected and interdependent but inactive Mcm2-7 complexes within the double hexamer.

Two explanations for unwinding by head-to-head CMG-Mcm10s are illustrated in *Figure 4A*. One process involves strand shearing (*Figure 4A*, left arrow) as observed in the reactions using the T-substrate in which the two non-homologous arms are sheared apart (*Figures 2* and *3*). An alternative process is that once sufficient dsDNA is unwound, the non-tracking strand may be expelled from the inner chamber of CMG, enabling completion of unwinding by conventional CMG helicase activity in which the CMGs encircle and translocate 3′−5′ on ssDNA (*Figure 4A*, right arrow).

In an effort to address whether strand expulsion occurs during unwinding, we constructed a modified T-substrate in which the two 50 bp non-complementary arms of the T-substrate are annealed to separate oligos with 5′ flaps, allowing us to monitor unwinding of each arm independently of the other (*Figure 5A*). This is in contrast to the substrate in *Figure 2* where both T-arms had to be melted in order to observe unwinding (see *Figure 5—figure supplement 1*). For the substrate in *Figure 5*, the 5′ flap in the 'C' oligo on the upper arm of the substrate (*Figure 5A*, left) was created by using a reverse polarity 3′−3′ linkage near the base of the flap to prevent CMG loading onto what would otherwise be a 3′ flap (*Figure 5—figure supplement 2*). The other 5′ flapped oligo, on the lower arm, is referred to here as the 'D' oligo. For these reactions, two trap oligos are used to prevent unwound products from reannealing: [trap]E prevents unwound oligo C from reannealing to oligo B while [trap]A is an unlabeled version of oligo A that prevents unwound oligos B and/or D from

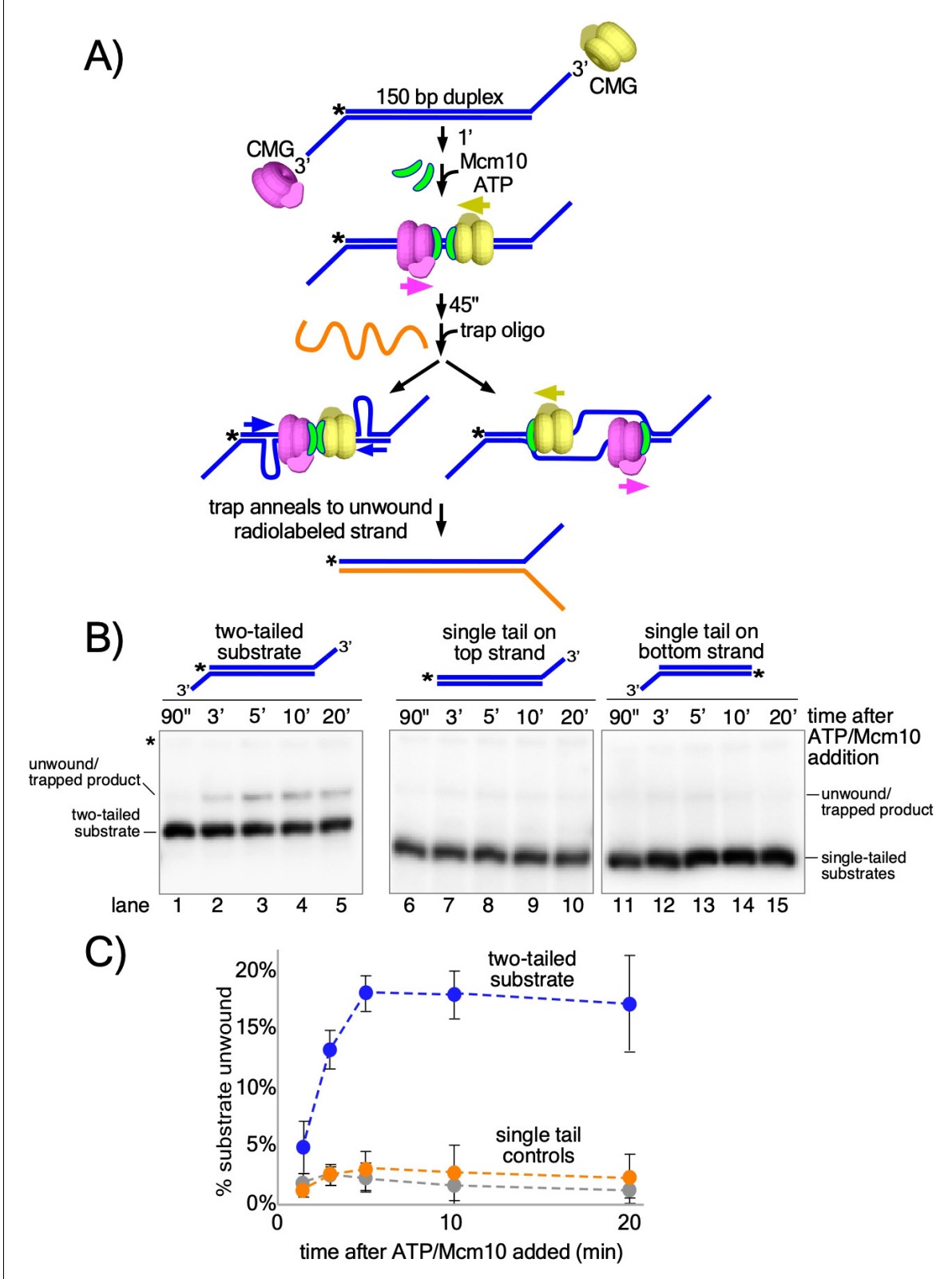

**Figure 4.** Head-to-head CMGs unwind dsDNA. (A) Scheme of the reaction with the double tailed substrate, and two possible processes that could result in unwinding the DNA. The color scheme is the same as in *Figure 1*. CMG is mixed with the substrate on ice and then incubated at 30° C in the absence of ATP (top) to allow the reaction to reach temperature. 1' later, ATP is added to allow CMGs to load onto the duplex in opposite orientations and block each other's progress (middle). At the same time as ATP, Mcm10 is added to initiate the duplex unwinding reaction. 45' later, an ssDNA trap

*Figure 4 continued on next page*

*Figure 4 continued*

oligo (orange) is added that quenches further CMG loading (*Figure 4—figure supplement 1*) and also anneals to the unwound radiolabeled product, creating a forked structure (bottom) that migrates at a distinct position in the gel from the substrate. Unwinding may occur either by: strand shearing (left arrow), or by strand expulsion (right arrow) to form CMGs that encircle ssDNA for conventional helicase activity. (B) Native PAGE gel time course of results using CMG + Mcm10. Lanes 1–5 show the reaction using the two-tailed substrate described in (A) while lanes 6–10 and 11–15 show control reactions using the same duplex with only a single tail at one end or the other. The migration of the substrates and unwound/trapped product are indicated to the left and right of the gel. (C) A plot of the data from (B) shows the averages of three independent trials using these substrates. The error bars show the standard deviations. See also *Figure 4—figure supplements 2–3*.

DOI: https://doi.org/10.7554/eLife.46515.010

The following figure supplements are available for figure 4:

**Figure supplement 1.** Addition of oligo trap prevents further loading of CMGM onto the two-tailed substrate.

DOI: https://doi.org/10.7554/eLife.46515.011

**Figure supplement 2.** CMG alone does not unwind the origin-duplex substrate.

DOI: https://doi.org/10.7554/eLife.46515.012

**Figure supplement 3.** Unwinding of the origin-duplex substrate in *Figure 4* requires ATP and CMG.

DOI: https://doi.org/10.7554/eLife.46515.013

reannealing to radiolabeled oligo A (*Figure 5A*, right). This experimental design enables visualization of all possible product outcomes of CMGM translocation on dsDNA, including whether CMG continually encircles dsDNA or whether it might expel one strand to encircle only the tracking strand at some point in the process (see *Figure 5A,D*).

When CMG loads onto the initial duplex segment of the modified substrate it can follow one of two pathways. If CMG translocates exclusively on duplex DNA during the reaction and melts both non-homologous arms, then both flapped oligos (oligos C and D in *Figure 5A*, middle right) will be released by the action of the CMG motor. Alternatively, it is possible that CMG will melt only oligo D (*Figure 5A*, top right). For example, if CMG initially translocates on dsDNA and melts a portion of the non-homologous arms and then expels the non-tracking strand, only the lower flapped oligo D will be unwound (*Figure 5A,D*). This same outcome could also arise if CMG remains on dsDNA but is faster in melting oligo D compared to oligo C (*Figure 5A,E*). Alternatively, despite the lack of a 5' flap on the initial flush duplex segment, CMG might take advantage of thermal fraying at the flush end and unwind the 3'-tailed oligo ($^{32}$P-oligo A in *Figure 5A*, bottom right) from the rest of the substrate. Previous experiments with similar substrates indicate that unwinding of flush duplexes is negligible (*Langston and O'Donnell, 2017*). By radiolabeling the 3'-tailed 'A' oligo, these potential products are observed as separate bands in a native gel (*Figure 5B*). The reaction conditions are identical to those of the experiments in *Figure 2C* (CMG + Mcm10) except for the modified substrate design and the use of two traps to prevent unwound substrates from reanealing (see *Figure 5A*, right).

The results of the experiment with the modified substrate reveal that melting of oligo D is the most prominent product of the reaction, appearing as a clear and distinct band in the gel as early as 90' after starting the reaction (band labeled *ABC in *Figure 5B*) and occurring on 40% of substrates over the full time course (*Figure 5C*). In contrast, the product expected for CMG encircling dsDNA and melting both arms simultaneously was ~10% of total substrate (*AB in *Figure 5B,C*), as observed in *Figure 2* for the 60 bp arm substrate. Continuous ssDNA translocation during the entire experiment was a minor outcome (~5%, *A in *Figure 5B,C*) as expected because there is no 5' flap on the initial duplex segment, further supporting the conclusion that CMG moves smoothly from ssDNA onto a flush duplex without unwinding it, as previously observed (*Langston and O'Donnell, 2017*).

As discussed above, the melting of oligo D may be explained either by CMG +Mcm10 expelling the non-tracking strand (*Figure 5D*) or by CMG +Mcm10 continuously encircling dsDNA but melting oligo D faster than oligo C (*Figure 5E*). While we cannot distinguish between these two mechanisms, we note that an ssDNA gate was previously demonstrated for *Drosophila* CMG (*Ilves et al., 2010*; *Moyer et al., 2006*), and subsequently shown to generalize to human CMG and *S. cerevisiae* CMG (*Kang et al., 2012*; *Langston and O'Donnell, 2017*). Also, in recent single molecule studies, we show that CMGMs act as individual particles and that CMGM on dsDNA, upon encountering an RPA-coated fork, expels one strand and transitions to encircle ssDNA where it moves slowly to unwind the duplex (*Wasserman et al., 2018*). Regardless, the ensemble experiments of *Figure 5*

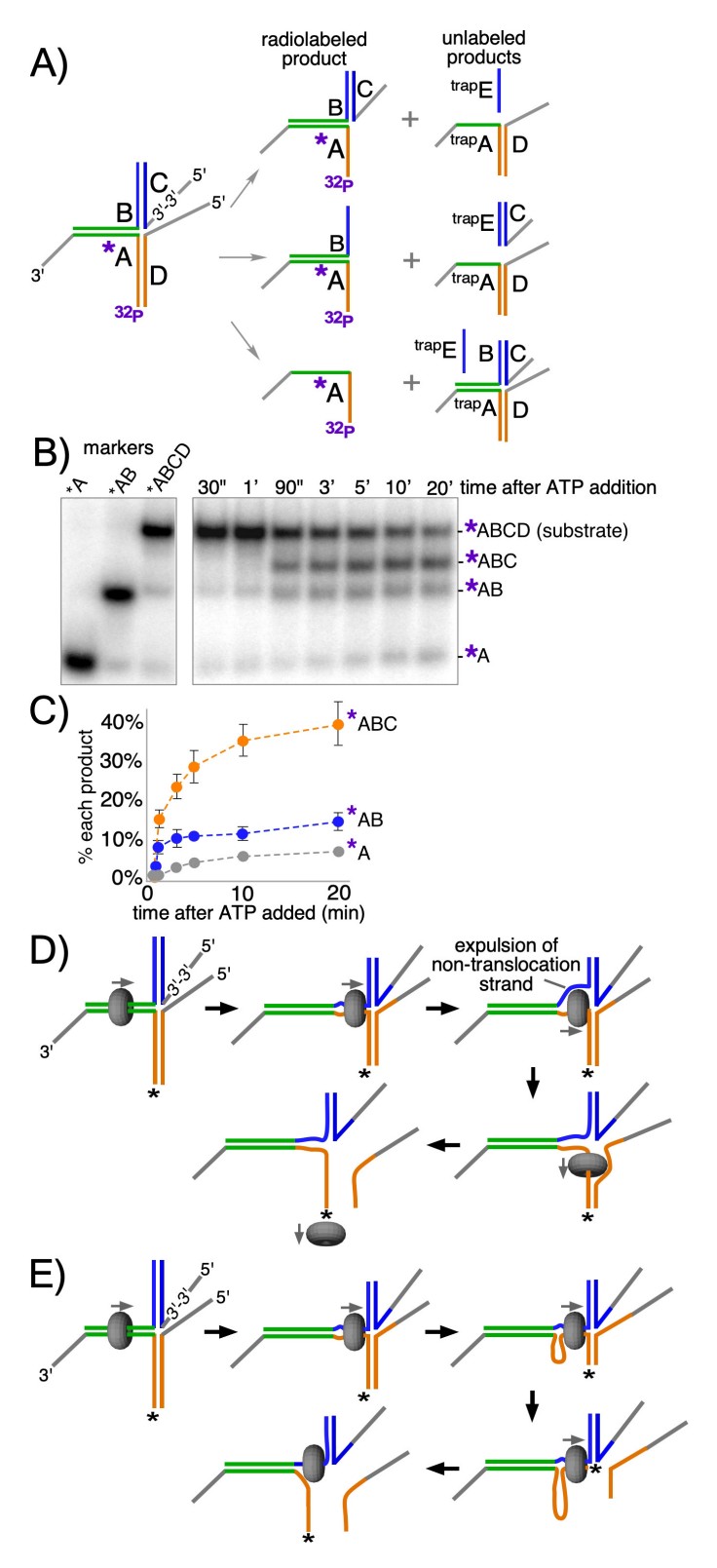

**Figure 5.** CMG + Mcm10 preferentially unwinds the tracking arm of a T-substrate. (**A**) Scheme of the substrate and possible reaction products depending on the pathway of unwinding. Oligo A is radiolabeled at the 5' end ($^{32}$P) and oligo C contains a reverse polarity (3'−3') linkage to create a 5' flap that prevents CMG loading onto oligo C (*Figure 5—figure supplement 2*). Two traps are used in the reaction, each in 40-fold excess over the substrate: one is unlabeled oligo A ($^{trap}$A, green and orange), which prevents unwound oligos D or B from re-annealing to unwound $^{32}$P-oligo A; the

*Figure 5 continued on next page*

*Figure 5 continued*

other is oligo E (^trapE, blue), which is complementary to oligo C and thus prevents unwound oligo C from re-annealing to oligo B. These traps are shown annealed to the possible products (right). (B) Representative native PAGE gel of the assay time course using CMG + Mcm10. The migration positions of the indicated DNA makers are shown at the left and the time course of the reaction is on the right. (C) Plot of the data from (B). The values shown are the average of three independent experiments and the error bars show one standard deviation. (D) Scheme of the strand expulsion pathway. CMG partially unwinds the two heterologous arms, creating ssDNA (top middle) and then the non-tracking strand B is expelled to the exterior of the ring (top right), followed by CMG encircling ssDNA (bottom right) and finishing unwinding of the lower flapped arm (bottom left). (E) Schematic of the DNA shearing pathway. CMG melts both arms (top middle) but at unequal rates (top right) such that the lower arm with the tracking strand is fully unwound before the upper arm (bottom right), leaving CMG encircling the non-tracking ssDNA (bottom left). See also *Figure 5—figure supplement 1*.

DOI: https://doi.org/10.7554/eLife.46515.014

The following figure supplements are available for figure 5:

**Figure supplement 1.** Description of why strand expulsion would not be observed using the substrate in *Figure 2*.

DOI: https://doi.org/10.7554/eLife.46515.015

**Figure supplement 2.** CMG+Mcm10 does not load onto the reverse polarity 5' flap.

DOI: https://doi.org/10.7554/eLife.46515.016

cannot rigorously distinguish whether CMG +Mcm10 expels the non-tracking strand or continuously encircles dsDNA to melt oligo D to form the ABC product.

## Discussion

The results presented in this report demonstrate that CMG + Mcm10 translocates on dsDNA with sufficient force to melt at least 60 bp (*Figure 2*), and that head-to-head CMGs can melt up to 150 bp (*Figure 4*). The head-to-head CMGs can be added individually and thus are uncoupled and do not need to be intimately connected as in the Mcm2-7 double hexamer. The experiments further demonstrate that Mcm10 is required for these CMG actions and that no other origin firing factors are needed. CMGM tracking on dsDNA with force preferentially interacts with the 3'−5' strand more than the 5'−3' strand, unlike the inactive Mcm2-7 double hexamer that does not unwind DNA and has an equal number of contacts with each strand of duplex DNA (*Abid Ali et al., 2017*; *Noguchi et al., 2017*). DNA melting by head-to-head CMGs directed against one another is expected to require a force of 65 pN, the force demonstrated to melt dsDNA in single-molecule experiments with optical traps attached to the ends of opposite strands of a linear duplex (*King et al., 2016*; *van Mameren et al., 2009*). Generation of 50–60 pN force has precedent in the ring-shaped FtsK segregation motor (*Pease et al., 2005*) and in phage packaging motors (*Smith et al., 2001*). In the case of origin melting, two CMG + Mcm10 are utilized and thus each CMG would only need half the 65 pN required for the ds-to-ssDNA phase transition.

DNA melting at an origin is a necessity for replication in all cell types, but occurs in different ways in bacteria and eukaryotes (reviewed in *Lyubimov et al., 2011* and *Parker et al., 2017*). While bacterial origin binding proteins melt enough DNA for two hexameric helicases to assemble directly onto ssDNA, eukaryotic ORC (Origin Recognition Complex) does not unwind DNA and instead conspires with many other factors to assemble two CMG helicases on dsDNA that are oriented head-to-head (*Bell and Labib, 2016*). Each CMG is demonstrated to untwist about 0.7 turns apiece (*Douglas et al., 2018*), but how the CMGs lead to sufficient DNA melting for strand expulsion and pass one another has been unknown except that Mcm10 is required (reviewed in *Bell and Labib, 2016*).

While the studies of this report provide a mechanistic overview of the origin unwinding step, additional details of this step remain to be determined. Mainly, we do not know the basis by which Mcm10 provides CMG the ability to unwind duplex while encircling duplex DNA. One possibility is that Mcm10 prevents slippage or backtracking by CMG, as backtracking in a ring shaped helicase has been documented previously (*Sun et al., 2011*). Indeed, recent studies show that the two CMGs at an origin can drift apart in buffer lacking ATP, indicating they are no longer tightly attached as in the Mcm2-7 double hexamer and are capable of backward slippage (*Douglas et al., 2018*). If Mcm10 were to prevent CMG backtracking it might more efficiently utilize the intrinsic force generated by ATP-driven CMG translocation. We do not know if Mcm10 acts on CMG at origins the same way as it acts upon CMG at replication forks. Unfortunately, CMG-Mcm10 complex with or without a

replication fork shows no additional Mcm10 density in cryoEM suggesting Mcm10 flexibility on CMG or multiple conformers of Mcm10 on CMG (*Douglas et al., 2018*; *Mayle et al., 2019*), and therefore additional studies will be needed to understand how Mcm10 acts upon CMG. For example, as an alternative to preventing backtracking, Mcm10 may induce a change in DNA structure that makes it easier to melt, as observed in monomeric helicases that orient the ssDNA and dsDNA at nearly right angles (*Gao et al., 2019*; *Lee and Yang, 2006*). In practice, the challenge to studying this process is that once two CMGs are formed, Mcm10 binding to the CMGs does not form a stable intermediate to examine because the CMGs quickly pass one another and form replication forks. The structure of CMG-Mcm10 intermediate species at an origin must await further studies. Hence, we propose below a plausible mechanism for Mcm10-dependent origin melting by CMG that utilizes the facts as we know them thus far.

We have shown previously that the N-terminal zinc fingers of CMG encircle and interact with both strands of duplex DNA just prior to the split point and that the duplex proceeds nearly straight into CMG before the leading strand enters the central channel (*Georgescu et al., 2017*; *O'Donnell and Li, 2018*). Furthermore, observations of Mcm2-7 head-to-head double hexamers on DNA show they are nearly in-line with one another while they encircle dsDNA (*Abid Ali et al., 2017*; *Li et al., 2015*; *Noguchi et al., 2017*; *Remus et al., 2009*). Thus we propose a model in *Figure 6* for origin melting assuming two in-line head-to-head CMGs (+Mcm10) at an origin. The length of the central channel of each CMG is about 110 Å which can enclose 20–30 bp of dsDNA. Thus, significant melting is required to provide ssDNA of sufficient length for one strand to be excluded from the central channel to the outside of the CMG particle. The motors of CMG are within the C-terminal domains and are separated by the N-terminal domains in a head-to-head orientation (*Figure 6*). Recent studies show that each CMG can untwist about 7 bp without Mcm10, probably within the C-terminal domains containing the ATP sites, but whether this is unwound DNA or untwisted DNA is not certain (*Douglas et al., 2018*). While this is insufficient for the dsDNA-to-ssDNA transition, it may be expected to facilitate the transition. Like other ring-shaped oligomers that encircle dsDNA (*Aathavan et al., 2009*; *Simonetta et al., 2009*), if CMG tracks primarily on one strand of dsDNA, as indicated by data of *Figure 3*, the dsDNA will come under tension, each strand being pulled in the opposite direction. Upon reaching the threshold tension for melting, the dsDNA between the CMGs will be sheared to ssDNAs and further tracking will produce additional ssDNA (*Figure 6*). The disposition of this sheared DNA is not clear, but one possibility is that the ssDNA might be stored within the N-tier by a conserved MCM-ssDNA binding motif, as revealed in a structure of the N-tier of an archaeal MCM bound to ssDNA (*Froelich et al., 2014*). Alternatively, DNA melting may occur within individual CMG-Mcm10 molecules, leaving DNA between them as duplex. These various possible intermediate states are indicated by the question mark in *Figure 6*.

Once sufficient DNA is unwound, the CMGs must expel one strand of ssDNA from the central channel to become bone fide helicases that can pass one another to initiate two divergent replication forks. Expulsion of a ssDNA implies a ssDNA gate within CMG. A ssDNA gate intrinsic to CMG is consistent with the demonstrated ability of CMG to self-load onto circular ssDNA for removal of 5' flap oligonucleotides bound to the circular DNA, demonstrated for *Drosophila* CMG and later for human and budding yeast CMG (*Ilves et al., 2010*; *Kang et al., 2012*; *Langston et al., 2014*). Turns in the ssDNA strands that remain after melting the dsDNA would not encumber these transitions because the turns can diffuse into the flanking dsDNA and be removed by topoisomerases (*Postow et al., 1999*). For example, the two CMG-Mcm10 complexes may rotate in opposite directions relative to one another, following the contour of the DNA (*Figure 6*), or the DNAs may rotate while the head-to-head CMG-Mcm10's remain stationary. In either case, the turns in the melted DNA will be pushed out the C-face of each CMG-Mcm10 complex to yield supercoils for topoisomerase action. Once the CMGs pass one another, they have been demonstrated to recruit Pol α-primase to form primed sites that, upon extension, become leading strands (*Aria and Yeeles, 2019*).

Despite the knowledge gained by the current report, it will require further study to understand the exact details in these last steps in origin initiation in which the CMGs unwind DNA while encircling dsDNA and transition to encircling ssDNA for bidirectional replication. Not only do we require further information on the exact role of Mcm10, but the presumed ssDNA gate has yet to be identified. In fact, many of the preceding steps leading up to assembly of CMG are still not characterized in molecular detail. These include the structural transitions of Mcm2-7 that occur upon DDK phosphorylation that enable it to bind Sld3/7 and Cdc45; the structure of the Mcm2-7-Sld3/7-Cdc45

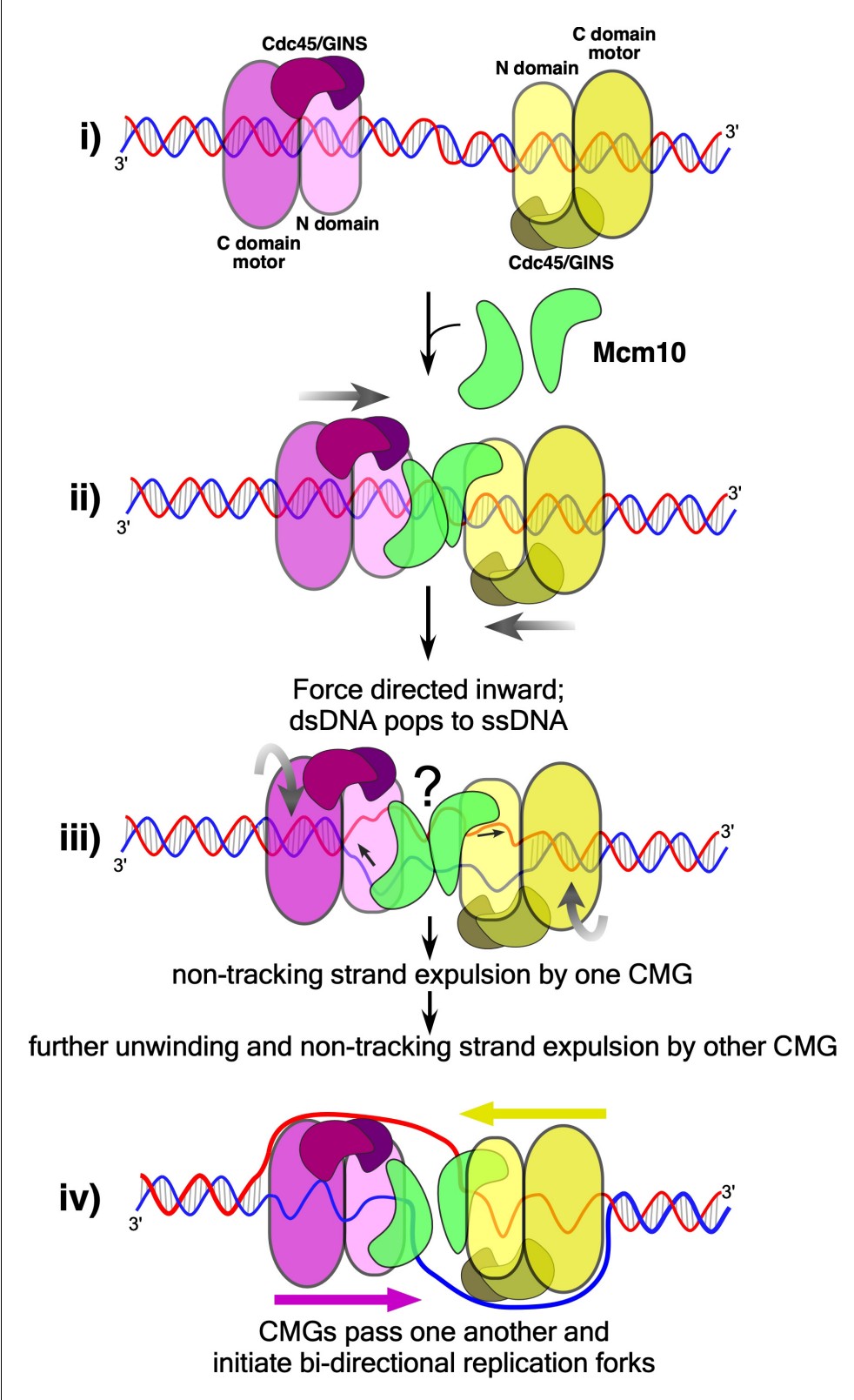

**Figure 6.** Model of CMG action at an origin. (i) Two head-to-head CMGs (purple and yellow) are formed at an origin surrounding dsDNA (blue and red). (ii) Addition of Mcm10 (green) promotes movement of the two CMGs toward one another (gray arrows). (iii) Upon colliding, the two CMG motors continue to exert force on their respective tracking strands and may continue to rotate (curved gray arrows), placing tension on the dsDNA between them and causing it to unwind. The presence of Mcm10 promotes significant shearing and melting of dsDNA. Small black arrows show the direction of

*Figure 6 continued on next page*

*Figure 6 continued*

DNA strand movement through the respective CMGs. While ssDNA is illustrated both inside and between CMGMs, it remains possible that DNA unwinding occurs within the N-and C-tiers of individual CMGs or some other process. The question mark reflects this uncertainty. (iv) The unwinding of dsDNA by CMGM observed in the current study provides a sufficient length of ssDNA for CMGs to expel the non-tracking strand and thus transition from encircling dsDNA to ssDNA and track 3′−5′ on their respective ssDNA past one another to form bidirectional forks (see also *Figure 1*).
DOI: https://doi.org/10.7554/eLife.46515.017

intermediate; and the structure of the Dbp11-Sld2-Sld3 complex and its interaction with GINS and Pol ε, all of which are required to form CMG from its component parts. Clearly, there is much more to be learned about origin initiation, and given the conservation of the factors involved in higher eukaryotes, the processes discovered in *S. cerevisiae* will likely generalize.

## Materials and methods

### Reagents and proteins

Radioactive nucleotides were from Perkin Elmer and unlabeled nucleotides were from GE Healthcare. DNA modification enzymes were from New England Biolabs. DNA oligonucleotides were from Integrated DNA Technologies except for those with methylphosphonate linkages which were from Biosynthesis (Lewisville, TX). CMG and Mcm10 were overexpressed and purified as previously described (*Georgescu et al., 2014*; *Langston et al., 2017*; *Langston et al., 2014*). Protein concentrations were determined using the Bio-Rad Bradford Protein stain using BSA as a standard.

### DNA substrates

For all radiolabeled oligonucleotides, 10 pmol of oligonucleotide was labeled at the 5′ terminus with 0.05 mCi [γ-$^{32}$P]-ATP using T4 Polynucleotide Kinase (New England Biolabs) in a 25 μl reaction for 30′ at 37°C according to the manufacturer's instructions. The kinase was heat inactivated for 20′ at 80°C.

For annealing, 4 pmol of the radiolabeled strand was mixed with 6 pmol of unlabeled complementary strand(s), NaCl was added to a final concentration of 200 mM, and the mixture was heated to 90°C and then cooled to room temperature over a time frame of >1 hr. DNA oligonucleotides used in this study are listed in *Supplementary file 1*.

The T20 substrate in *Figure 2* was made by annealing unlabeled oligos T20A and T20B to radiolabeled T20C. Likewise, the T30 substrate was made by annealing unlabeled T30A and T30B to radiolabeled T30C; and the T60 substrate was made by annealing unlabeled T60A and T60B to radiolabeled T60C. See *Supplementary file 1* for oligo sequences.

The substrates in *Figure 3B* were made exactly as the T30 substrate in *Figure 2* except that oligo 'T30B MP' was substituted for T30B for the substrate with methylphosphonates in the 5′−3′ strand and oligo 'T30A MP' was substituted for T30A for the substrate with methylphosphonates in the 3′−5′ strand. See *Supplementary file 1* for oligo sequences.

The substrates in *Figure 4B* were made by annealing unlabeled 'ORI Bottom 3′ tail' to radiolabeled 'ORI Top 3′ tail' (lanes 1–5); unlabeled 'ORI Bottom no tail' to radiolabeled 'ORI Top 3′ tail' (lanes 6–10); and unlabeled 'ORI Top no tail' to radiolabeled 'ORI Bottom 3′ tail' (lanes 11–15). See *Supplementary file 1* for oligo sequences.

The substrate in *Figure 5* was made in two stages. First, unlabeled T50B was annealed to radiolabeled T50A as indicated above. Then T50C and T50D were added and the mixture was incubated 10′ at 30°C to form the full T50ABCD substrate.

### DNA unwinding assays

Unless otherwise noted, all reactions were performed at 30°C and contained (final concentrations) 20 nM CMG and 40 nM Mcm10 with 0.5 nM radiolabeled DNA substrate in a buffer consisting of 20 mM Tris Acetate pH 7.6, 5 mM DTT, 0.1 mM EDTA, 10 mM MgSO$_4$, 20 mM KCl, 40 μg/ml BSA and 1 mM ATP.

For the T-substrate assays in *Figures 2* and *3*, CMG was pre-incubated with the DNA substrate for 10′ at 30°C in the absence of ATP and the reaction was started by addition of ATP and, where

indicated, Mcm10, in a final volume of 80 µl. To prevent re-annealing of the unwound radiolabeled DNA, an unlabeled version of the radiolabeled oligo (T20C, T30C or T60C depending on the substrate; see *Supplementary file 1* for oligo sequences) was added as a trap 1' after starting the reaction to a final concentration of 20 nM. At the indicated times, 12 µl reaction aliquots were removed, stopped with 4 µl STOP/LOAD buffer containing 0.1M EDTA, 5% SDS, 25% glycerol, and 0.01% each of xylene cyanol and bromophenol blue, and flash frozen in liquid nitrogen. Upon completion of the reaction, flash frozen reaction products were thawed quickly in water at room temperature and separated on 10% native PAGE minigels by electrophoresis at 100V for 75 min in TBE buffer. Gels were washed in distilled water, mounted on Whatman 3 MM paper, wrapped in plastic and exposed to a phosphor screen that was scanned on a Typhoon 9400 laser imager (GE Healthcare). Scanned gels were analyzed using ImageQuant TL v2005 software.

Reactions using the flapped substrate in *Figure 5* were performed as described for *Figure 2* except the reaction volume was 110 µl and two unlabeled trap oligos (T50A and T50E, see *Supplementary file 1*) were used to prevent re-annealing of any unwound substituents and preserve the three potential products of the reaction. Reaction aliquots (10 µl) were stopped at the indicated times by addition of 4 µl of buffer containing 150 mM EDTA/7% SDS. 1 µl Proteinase K was added and the mixture was incubated 10' at 30° C after which 3 µl STOP/LOAD buffer was added and the sample was flash frozen in liquid nitrogen. Upon completion of the reaction, frozen reaction products were thawed and processed as described above. Samples were separated on 5% native PAGE minigels by electrophoresis at 80V for 80 min in TBE buffer.

For the reactions in *Figure 4*, 40 nM CMG was pre-incubated with the DNA substrate for 1' at 30° C in the absence of ATP and then ATP was added to allow CMG to load onto the 3' tails and thread onto the duplex DNA substrate. The total reaction volume was 55 µl. Mcm10 was added along with ATP to a final concentration of 80 nM followed 45' later by addition of an unlabeled trap oligo ('ORI Bottom 5' tail' at 20 nM final, see *Supplementary file 1*) that binds to the unwound radiolabeled DNA and shifts it to a unique position in the native PAGE gel. For the reaction in lanes 11–16, the trap was 'ORI Top 5' tail' (*Supplementary file 1*). At the indicated times following ATP/Mcm10 addition, 10 µl reaction aliquots were removed and stopped by addition of 4 µl of buffer containing 150 mM EDTA/7% SDS. 1 µl Proteinase K was added and the mixture was incubated 10' at 30° C after which 3 µl STOP/LOAD buffer was added and the sample was flash frozen in liquid nitrogen. Upon completion of the reaction, frozen reaction products were thawed and processed as described above for the reactions of *Figure 2* except that electrophoresis was at 100V for 120 min.

## Acknowledgements

The authors wish to thank Daniel Zhang for purification of CMG and Nina Yao for artwork in *Figure 1*.

## Additional information

### Funding

| Funder | Grant reference number | Author |
| --- | --- | --- |
| Howard Hughes Medical Institute | | Michael E O'Donnell |
| National Institutes of Health | GM115809 | Michael E O'Donnell |

The funders had no role in study design, data collection and interpretation, or the decision to submit the work for publication.

### Author contributions

Lance D Langston, Conceptualization, Resources, Formal analysis, Investigation, Writing—original draft, Writing—review and editing; Michael E O'Donnell, Conceptualization, Formal analysis, Supervision, Funding acquisition, Writing—original draft, Project administration, Writing—review and editing

## Author ORCIDs

Lance D Langston ⬤ https://orcid.org/0000-0002-2736-9284
Michael E O'Donnell ⬤ https://orcid.org/0000-0001-9002-4214

## Decision letter and Author response

Decision letter https://doi.org/10.7554/eLife.46515.021
Author response https://doi.org/10.7554/eLife.46515.022

## Additional files

### Supplementary files

• Supplementary file 1. DNA oligonucleotides used in this study.
DOI: https://doi.org/10.7554/eLife.46515.018

• Transparent reporting form
DOI: https://doi.org/10.7554/eLife.46515.019

### Data availability

All data generated or analysed during this study are included in the manuscript and supporting files.

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
