## [Decision Letter]

Thank you for submitting your article "An explanation for origin unwinding in eukaryotes" for consideration by *eLife*. Your article has been reviewed by two peer reviewers, and the evaluation has been overseen by a Reviewing Editor and Cynthia Wolberger as the Senior Editor. The reviewers have opted to remain anonymous.

The reviewers have discussed the reviews with one another and the Reviewing Editor has drafted this decision to help you prepare a revised submission.

Summary:

The present paper investigates the mechanism of origin unwinding by the *S. cerevisiae* CMG helicase and role of MCM10 using a number of clever biochemical experiments. The major conclusions drawn from these studies are that the CMG tracks mainly along the 3'-5' strand of dsDNA to unwind the strands, and that MCM10 helps the CMG in creating added force to unwind a long stretch of dsDNA, which in turn enables the CMG to switch from dsDNA to ssDNA. While some of these conclusions are well supported by experiments, others such as CMG switching strands and added force production by MCM10 require additional considerations.

Essential revisions:

1) There is no direct evidence that MCM10 provides additional force to unwind the DNA per se. Figure 2 shows that CMG unwinds the T-junction substrate with a higher efficiency in the presence of MCM10, and from these results, it is concluded that CMG+MCM10 translocates with more force than CMG alone while encircling dsDNA. This is an intellectual jump, as the assay simply shows an increase in the yield of the unwound products, which could simply be due to more active or stable CMG and DNA complexes in the presence of MCM10. Controls should be performed to rule out this possibility. The authors should also elaborate what is meant by force production. Is it that the processivity of CMG is greater with MCM10, or is it that the ATPase is more efficient, translocation along DNA is faster, etc.? If controls cannot be done then force production explanation should be toned down and other possible explanations provided.

2) Figure 4 shows that the CMG and MCM10 can unwind a 150 bp dsDNA with two ssDNA tails and the cartoon shows that this happens with two opposing MCMs. This may be misleading. There is no evidence that only two MCMs are loaded on the DNA and not an array of MCMs. Also, the cartoon shows that unwinding happens by a strand switching mechanism. However, opposing MCMs could unwind without strand switching by pulling the two DNA strands in the opposite 3'-5' directions to shearing the base pairs. Evidence should be provided that only two CMGs are loaded on the 150 bp duplex DNA under the conditions of the experiment, as illustrated in the figure, otherwise the figure and the interpretation should be revised.

3) Figure 5 shows that CMG unwinds a broken T-junction substrate to produce mainly the ABC product. This is interpreted to be due to strand switching, but again MCM might be able to pull in the 3'-5 direction on strand A (as shown in Figure 2A) and since C and D are not linked, the reaction will unwind strand D without strand switching. For example, the helicase might pull on the 3'-5' strand while still encircling dsDNA and catalyzing unwinding, which would cause the yellow colored DNA arm in the 3'-3' experiment to give the ABC (strand switch) product. It's unclear whether there is a straightforward experiment to prove strand switching, but the authors should discuss alternative ways in which their substrates could be unwound in the assay that does not involve such a mechanism.

By way of illustration, consider the effects of the addition of trap C (as mentioned in the Results), which could give rise to the product expected from the strand switching model (see the attached annotated copy of the model for clarification). This experiment should be run and evaluated without trap C. Also, please indicate, within the figures in the main text, which traps were used in the experiments and how much of them, and what final products would be formed from trap annealing to the unwinding products.

4) Regardless whether all of the requested controls can be performed, then the phrase 'strand switching' is probably not the most accurate way to describe the authors' preferred process. The model as depicted doesn't show any change in the strands that the CMG translocates along – it's always the same 3'-5' strand (e.g., 'Watson'). To truly switch, the CMG would have to jump tracks to operate on 'Crick,' as has been proposed for SF1 and SF2 helicases based on certain some single molecule experiments. What the authors seem to be proposing is really 'strand expulsion', not switching. This phrasing should be clarified.

---

## [Author Response]

Essential revisions:1) There is no direct evidence that MCM10 provides additional force to unwind the DNA per se. Figure 2 shows that CMG unwinds the T-junction substrate with a higher efficiency in the presence of MCM10, and from these results, it is concluded that CMG+MCM10 translocates with more force than CMG alone while encircling dsDNA. This is an intellectual jump, as the assay simply shows an increase in the yield of the unwound products, which could simply be due to more active or stable CMG and DNA complexes in the presence of MCM10. Controls should be performed to rule out this possibility. The authors should also elaborate what is meant by force production. Is it that the processivity of CMG is greater with MCM10, or is it that the ATPase is more efficient, translocation along DNA is faster, etc.? If controls cannot be done then force production explanation should be toned down and other possible explanations provided.

We are grateful for this comment, and fully agree that addition of “force” to CMG by Mcm10 is not the only way to explain the data. We have shown earlier in an *eLife* publication that Mcm10 substantially enhances the processivity of CMG while acting as a helicase, and we have done additional experiments showing that Mcm10 enables CMG to displace lac repressors, but have not added this data to the manuscript because we realize that these results can also be explained by alternative means in the same way as Mcm10 stimulates the unwinding of longer heteroduplex arms of the T-substrate. In overview, we realize that biochemical experiments will not enable a conclusion about the force with which CMG, or CMGM acts. This would require a direct force measurement by a biophysical method, such as single molecule measurements that can directly measure force.

We have therefore revised the manuscript by toning down the term “force”, and adding new explanations, as advised by the review. These include that Mcm10 may enhance the affinity of CMG for the DNA, and also that Mcm10 may prevent CMG backtracking on DNA. These alternate explanations are all equally likely until experiments are performed to distinguish among them.

2) Figure 4 shows that the CMG and MCM10 can unwind a 150 bp dsDNA with two ssDNA tails and the cartoon shows that this happens with two opposing MCMs. This may be misleading. There is no evidence that only two MCMs are loaded on the DNA and not an array of MCMs. Also, the cartoon shows that unwinding happens by a strand switching mechanism. However, opposing MCMs could unwind without strand switching by pulling the two DNA strands in the opposite 3'-5' directions to shearing the base pairs. Evidence should be provided that only two CMGs are loaded on the 150 bp duplex DNA under the conditions of the experiment, as illustrated in the figure, otherwise the figure and the interpretation should be revised.

We agree with the reviewers on these points and have performed additional experiments and thus revised the text and figure. The DNA shearing explanation is our original intention and as pointed out by the review is consistent with the data. Thus we have now included this explanation in both the text and the figure illustration. We have also revised the text to focus on the orientation of CMGs required to observe unwinding, and revise the manuscript to state that this may be performed by at least two CMGs on the DNA (i.e. especially upon comparing the single tail loading site controls).

Ring shaped helicases typically show low activity in unwinding assays because of the requirement to thread onto DNA ends, which leads to low template occupancy compared to SF1 and SF2 helicases. But to further limit the opportunity for > 2 CMGs to load on an individual DNA substrate, we made significant modifications to the experimental design in Figure 4. First, we eliminated the 10’ pre-incubation of CMG+substrate+ATP and replaced it with a 1’ pre-incubation of CMG+substrate without ATP. The reaction is now started by addition of ATP and Mcm10 and then 45” later we add a trap that shuts down additional CMG loading. We added a new supplementary figure (Figure 4—figure supplement 1) documenting that addition of the trap prior to addition of ATP/Mcm10 completely eliminates unwinding, indicating that the trap sequesters CMG in solution, preventing further loading. Thus, the window of opportunity for CMG to load onto the substrate is reduced from over 10’ in our previous experiments to 45 seconds in the current version. We also note that unwinding continues well beyond the point at which the trap is added, which indicates that the observed unwinding is attributable to CMG loaded in the 45” window rather than to continuous loading of additional CMGs. This is pointed out in the revised text.

3) Figure 5 shows that CMG unwinds a broken T-junction substrate to produce mainly the ABC product. This is interpreted to be due to strand switching, but again MCM might be able to pull in the 3'-5 direction on strand A (as shown in Figure 2A) and since C and D are not linked, the reaction will unwind strand D without strand switching. For example, the helicase might pull on the 3'-5' strand while still encircling dsDNA and catalyzing unwinding, which would cause the yellow colored DNA arm in the 3'-3' experiment to give the ABC (strand switch) product. It's unclear whether there is a straightforward experiment to prove strand switching, but the authors should discuss alternative ways in which their substrates could be unwound in the assay that does not involve such a mechanism.

We agree with the reviewers, and thank them for this interpretation. We originally thought that dividing the C-oligo into two oligos (C and D), would distinguish between strand shearing (of both arms) and strand expulsion (displacing oligo D only). But thanks to the reviewers we now realize the result we observed could be explained by DNA shearing, provided the oligo A-D arm was more rapidly sheared apart than the oligo B-C arm. Thus we have revised the text as advised. We note that if the ABC product is produced with no strand expulsion, then CMG would remain encircling the non-tracking B strand of the ABC DNA product (see revised Figure 5).

While we do not stand behind strand expulsion in this report, but focus on strand separation, we cite a BioRXiv study in collaboration with the Shixin Liu lab in which we have visually observed this step in a single-molecule approach. This work has now accepted to Cell (i.e. in press). We cite the BioRXiv study because the accepted Cell report will take significant time for publication. We also point out that a ssDNA gate in CMG can be inferred from early studies of *Drosophila* CMG melting a 5’ tailed oligo from circular ssDNA by the Botchan group (Ilves et al., 2010; Moyer et al., 2006), and subsequently shown to generalize to human CMG and *S. cerevisiae* CMG (Kang et al., 2012; Langston et al., 2014). This supports the existence of an ssDNA gate in CMG even though cryo EM structures show the Mcm2-7 ring is closed in CMG.

Since Figure 5 is consistent with, but does not prove strand expulsion, we have revised the manuscript to explain that the DNA construct with individual oligos for each arm can determine whether the tracking arm (oligo C displacement) is melted before the non-tracking arm (oligo D displacement). We provide two explanations for the observed result (in the text and in the figure illustration): 1) that a strand expulsion path may produce the ABC product and 2) that DNA melting akin to that of Figures 2, 3 may produce the ABC product if shearing is faster on one arm vs. the other, and that we cannot distinguish between the two. Because the illustration in Figure 5D is now part of the main figure, we have eliminated Supplementary Figure 11 which illustrated the same reaction.

By way of illustration, consider the effects of the addition of trap C (as mentioned in the Results), which could give rise to the product expected from the strand switching model (see the attached annotated copy of the model for clarification). This experiment should be run and evaluated without trap C. Also, please indicate, within the figures in the main text, which traps were used in the experiments and how much of them, and what final products would be formed from trap annealing to the unwinding products.

We very much appreciate this comment because it points out to us that we did not clearly explain the traps that we used, and we did not show them in the figure. Indeed, no one could have determined this from the original figures. The traps are unlabeled oligos that hybridize to oligos C and D, the split oligo arms. These are now shown in the figure and are named oligos ^trap^A (which is oligo A not labeled with ^32^P) and oligo ^trap^E in the figure. ^trap^E was confusingly called “T50C trap” in the original manuscript and we have changed the name of this oligo to “T50E” in Supplementary file 1 to reflect the fact that the trap used to prevent reannealing of Oligo C is distinct from (and complementary to) T50C.

We have now revised the text and figure illustrations. In the text we clarify the reaction and traps, and the revised figure illustrations now show the traps bound to the expected products. We appreciate the opportunity to have clarified this.

4) Regardless whether all of the requested controls can be performed, then the phrase 'strand switching' is probably not the most accurate way to describe the authors' preferred process. The model as depicted doesn't show any change in the strands that the CMG translocates along – it's always the same 3'-5' strand (e.g., 'Watson'). To truly switch, the CMG would have to jump tracks to operate on 'Crick,' as has been proposed for SF1 and SF2 helicases based on certain some single molecule experiments. What the authors seem to be proposing is really 'strand expulsion', not switching. This phrasing should be clarified.

We completely agree that use of the term “strand switching” (i.e. our lab jargon) misrepresented the event that we were meaning to refer to, and that “strand expulsion” is the more accurate term for what we meant to convey. We have revised the manuscript throughout, to omit the term “strand switching” and instead use “strand expulsion”. As mentioned above, we no longer stand behind strand expulsion as being unambiguously demonstrated in this paper, and instead focus on the dsDNA melting aspect of CMG+Mcm10 while encircling dsDNA. We have directly shown a ssDNA gap in CMG for strand expulsion in a single molecule paper accepted for publication in *Cell* and refer to the bioRxiv citation of our single-molecule paper on this subject since we expect that the *Cell* paper will be published after this *eLife* report.